# Hierarchical Graph Signal Processing for Collaborative Filtering

## ABSTRACT

Graph Signal Processing (GSP) has proven to be a highly effective and efficient tool for predicting user future interactions in recommender systems. However, current GSP methods recognize user interaction patterns based on the interactions of all users, so that the recognized interaction patterns are not fully user-matched and easily impacted by other users with different interaction behaviors, resulting in sub-optimal recommendation performance. To this end, we propose a hierarchical graph signal processing method (HiGSP) for collaborative filtering, which consists of two key modules: 1) the cluster-wise filter module that recognizes user unique interaction patterns merely from interactions of users with similar preferences, making the recognized patterns able to reflect user preference without being influenced by other users with different interaction behaviors, and 2) the globally-aware filter module that serves as a complementary to the cluster-wise filter module to recognize user general interaction patterns more effectively from all user interactions. By linearly combining these two modules, HiGSP can recognize user-matched interaction patterns, so as to model user preference and predict user future interactions more accurately. Extensive experiments on six real-world datasets demonstrate the superiority of HiGSP compared to other GCN-based and GSP-based recommendation methods in terms of efficacy and efficiency.

## KEYWORDS

User preference modeling, graph signal processing

**ACM Reference Format:**
Anonymous Author(s). 2018. Hierarchical Graph Signal Processing for Collaborative Filtering. In *Proceedings of Make sure to enter the correct conference title from your rights confirmation email (Conference acronym 'XX)*. ACM, New York, NY, USA, 11 pages. https://doi.org/XXXXXXX.XXXXXXX

## 1 INTRODUCTION

Graph Signal Processing [7, 20], which is an extension of signal processing theory [19] on graph data, has recently received increasing attention and become a powerful tool in recommender systems due to its promising accuracy and parameter-free characteristic [15, 22]. By constructing various types of filters (e.g., static filter [22], dynamic filter [31]) on user interaction signals, it can extract different types of interaction patterns, i.e., item transition patterns, for user preference modeling, and thereby accurately predict user future interactions.

Figure 1: A toy example of interactions between four users and four items, where 1 indicates a user-item interaction, and 0 otherwise.

Existing methods [15, 22] design filters over the whole user interactions, so they mainly focus on recognizing user general interaction patterns to model user preference. However, merely using user general interaction patterns will lead to the deviation in user preference modeling, since the recognized general interaction patterns are not fully user-matched and easily impacted by other users with different interaction behaviors, thereby resulting in sub-optimal prediction of user future interactions. Figure 1 shows a toy example containing the interactions of four users. We can observe that different users have different interaction behaviors, Alice and Bob tend to interact with items related to music, while Jim and Tom prefer items related to electronic products. Generally, user general interaction patterns recognized by existing methods [15, 22] integrate four users' interaction behaviors. Therefore, when predicting the Alice's future interactions, her prediction results will be affected by the interaction behaviors of all other users (such as Jim and Tom) who have different preferences, thereby affecting the accuracy of her future interaction predictions.

To tackle this problem, we propose a hierarchical graph signal processing method HiGSP for collaborative filtering. User segmentation theory in the marketing field [6] points out that no two users are alike and none is entirely unique either. By segmenting users into distinct groups based on shared characteristics, we are able to better understand what motivates groups and users in groups. Inspired by this, we first cluster users based on their interactions, so that users within the same cluster have similar preferences, while users across different clusters have different preferences. Then, we design a cluster-wise filter module to recognize the interaction patterns of users in their respective clusters, thereby making the recognized interaction patterns unique, i.e., reflecting user preference without being influenced by other users with different interaction behaviors. In addition, we design a globally-aware filter module, which serves as a complementary to cluster-wise filter module, to recognize user general interaction patterns more effectively with the carefully devised high-order low-pass filter. Finally, we use a linear model to combine these two modules, so as to obtain user-matched interaction patterns, thereby making the user preference modeling and user future interactions prediction more accurate. We conduct extensive experiments on six real-world datasets, and the results show that HiGSP outperforms existing GCN-based and

GSP-based recommendation methods in terms of prediction accuracy, while achieves comparable results with the most efficient methods in terms of training efficiency. Besides, we conduct a visualization and a case study to further demonstrate the rationality and effectiveness of HiGSP.

Our main contributions are summarized as follows:

- We propose a cluster-wise filter module to recognize user unique interaction patterns without being influenced by other users with different interaction behaviors.
- We propose a globally-aware filter module, as a supplement to the cluster-wise filter module, to recognize user general interaction patterns more effectively.
- We conduct extensive experiments[1] to demonstrate the superiority of our proposed method compared to the state-of-the-art GCN-based and GSP-based recommendation methods.

## 2 PRELIMINARIES

### 2.1 Graph Signal Processing

*2.1.1 Graph Signal.* Generally, a graph $\mathcal{G} = (\mathcal{V}, \mathcal{E})$ with node set $\mathcal{V}$ and edge set $\mathcal{E}$ can be represented as an adjacency matrix $\mathbf{A} \in \{0, 1\}^{N \times N}$, where $N$ is the number of node, and if there is an edge between node $v_i$ and $v_j$, then $\mathbf{A}_{i,j} = 1$, otherwise $\mathbf{A}_{i,j} = 0$. The graph signal in essence is a mapping $f : \mathcal{V} \rightarrow \mathbb{R}$, and it can be represented as a vector $\mathbf{x} = [x_1, \cdots, x_N]^T$, where $x_i$ can be viewed as the signal strength on node $v_i$.

Graph laplacian matrix is an important and commonly used matrix in spectral graph theory [5, 23], and it is defined as $\mathbf{L} = \mathbf{D} - \mathbf{A}$, where $\mathbf{D} = \text{diag}(\mathbf{A1})$ is the degree matrix of graph $\mathcal{G}$. The normalized form of graph laplacian matrix can be expressed as $\tilde{\mathbf{L}} = \mathbf{I} - \tilde{\mathbf{A}}$, where $\tilde{\mathbf{A}} = \mathbf{D}^{-1/2}\mathbf{A}\mathbf{D}^{-1/2}$.

*2.1.2 Graph Filter and Graph Convolution.* As the normalized laplacian matrix $\tilde{\mathbf{L}}$ is a real and symmetric matrix, it can be decomposed into $\tilde{\mathbf{L}} = \mathbf{U}\mathbf{\Lambda}\mathbf{U}^T$, where $\mathbf{\Lambda} = \text{diag}(\lambda_1, \cdots, \lambda_n)$ is the eigenvalue matrix and $0 = \lambda_1 \leq \cdots \leq \lambda_n = 2$, $\mathbf{U}$ is the eigenvector matrix. Graph filter $\mathbf{F}$ is constructed upon the normalized laplacian matrix $\tilde{\mathbf{L}}$ with a frequency response function $f(\cdot)$

$$\mathbf{F} = f(\tilde{\mathbf{L}}) = \mathbf{U}\text{diag}(f(\lambda_1), \cdots, f(\lambda_n))\mathbf{U}^T. \tag{1}$$

The graph convolution of a given graph signal $\mathbf{x}$ is defined as

$$\mathbf{y} = \mathbf{F}\mathbf{x} = \mathbf{U}\text{diag}(f(\lambda_1), \cdots, f(\lambda_n))\mathbf{U}^T\mathbf{x}. \tag{2}$$

The graph signal $\mathbf{x}$ is first transformed from spatial domain to spectral domain through Graph Fourier Transform $\mathbf{U}^T$, then the undesired frequencies of the signal are removed by the filter in frequency domain. Finally, the clean signal is transformed back to spatial domain through inverse Graph Fourier Transform $\mathbf{U}$.

### 2.2 Notations

Let the user set be $\mathcal{U}$, and the item set be $\mathcal{V}$, and $|\mathcal{U}| = M, |\mathcal{V}| = N$. The interactions between users and items can be represented by a user interaction matrix $\mathbf{R} \in \{0, 1\}^{M \times N}$. If there is an interaction between user $u \in \mathcal{U}$ and item $v \in \mathcal{V}$, then $\mathbf{R}_{u,v} = 1$, otherwise $\mathbf{R}_{u,v} = 0$. The normalized interaction matrix can be

---

Linear Filter        Ideal Low-pass Filter

**Figure 2: The user interaction patterns, i.e., transitions between four items, recognized by linear filter and ideal low-pass filter based on the toy example in Figure 1.**

represented as $\tilde{\mathbf{R}} = \mathbf{D}_U^{-1/2}\mathbf{R}\mathbf{D}_I^{-1/2}$, where $\mathbf{D}_U^{-1/2} = \text{diag}(\mathbf{R1})$ and $\mathbf{D}_I^{-1/2} = \text{diag}(\mathbf{R}^T\mathbf{1})$ are the user and item degree matrices.

### 2.3 User Interaction Patterns

In this paper, we define user interaction patterns as the item transition matrices that are constructed from user historical interactions:

*Definition 2.1.* Given a user interaction matrix $\mathbf{R}$ and a mapping $g : \mathbb{R}^{M \times N} \rightarrow \mathbb{R}^{N \times N}$, user interaction pattern can be defined as an item transition matrix $\mathbf{F} = g(\mathbf{R})$, where the element $\mathbf{F}_{ij}$ describes the transition between item $i$ and item $j$, and it is proportional to probability that user interacts with item $j$ after interacting with item $i$ according to historical interactions.

Analogy to transition patterns of states in the system that can be recognized by transition matrix in Markov Chain, user interaction patterns can be recognized by graph filters in GSP, and different graph filters can recognize different user interaction patterns. For example, the linear filter $\mathbf{F} = \tilde{\mathbf{R}}^T\tilde{\mathbf{R}}$ recognizes the transition of items that share directly connected users as user interaction patterns, while the ideal low-pass filter $\mathbf{F} = \mathbf{D}_I^{-1/2}\bar{\mathbf{U}}\bar{\mathbf{U}}^T\mathbf{D}_I^{1/2}$ recognizes the transition of items that share directly and indirectly connected users as user interaction patterns, since the former is equivalent to a single-layer GCN, while the latter can be viewed as an infinite-layer GCN [22]. Figure 2 shows the user interaction patterns of the toy example in Figure 1, which are recognized by the linear filter and the ideal low-pass filter. We can find that the recognized user interaction patterns are different between different filters.

## 3 METHOD

### 3.1 Overview

In this section, we introduce our method HiGSP, which aims to more accurately model user preference from user interactions. Specifically, HiGSP consists of two important modules, a cluster-wise filter module used to recognize user unique interaction patterns, and a globally-aware filter module used to recognize user general interaction patterns. Figure 3 shows the workflow of HiGSP. Next, we introduce these modules in details and how to infer user future interactions with these modules.

### 3.2 Cluster-wise Filter Module

User segmentation theory in marketing field [6] highlights that segmenting users into different groups based on their shared characteristics is beneficial to understand user characteristics, so as to

---

[1]The code is released for review: https://anonymous.4open.science/r/HiGSP-F496/.

**Figure 3: The workflow of HiGSP, which adopts a cluster-wise filter module (the upper part, number of clusters $C = 2$, order $k_1 = 2$) and a globally-aware filter module(the lower part, number of primary components $K = 2$, order $k_2 = 2$) to make future interaction predictions for users. Note that we use user interaction patterns, i.e., item-item relationship graphs, to represent the filters. The color depth of the edges in all graphs reflects the strength of connection between nodes.**

develop personalized marketing strategies for users. Inspired by this theory, we propose to first cluster users based on their historical interactions, so that users in the same cluster have similar interaction behaviors, while users in different clusters have significantly different behaviors. By separating unrelated users based on their interaction behaviors before recognizing user interaction patterns, the recognized user interaction patterns will be unique and not be influenced by other users with different interaction behaviors, thereby accurately modeling user preference and predicting user future interactions.

In HiGSP, we use Mixture-of-Gaussian method [18] to cluster users, but HiGSP is orthogonal to the clustering method, so other clustering methods, such as BIRCH [37], can also be used to cluster. Suppose all users are divided into $C$ clusters, and the user interactions in the $c$-th ($1 \leq c \leq C$) cluster can be represented as $\mathbf{R}_c \in \mathbb{R}^{M_c \times N}$, where $M_c$ is the number of users in the $c$-the cluster. We design the following filter $\mathbf{F}_c$ to recognize the user interaction patterns in the $c$-the cluster

$$\mathbf{G}_c = \mathbf{R}_c^T \mathbf{R}_c, \quad \tilde{\mathbf{G}}_c = \mathbf{D}_c^{-\frac{1}{2}} \mathbf{G}_c \mathbf{D}_c^{-\frac{1}{2}}, \quad (3)$$

$$\mathbf{F}_c = \mathbf{I} - (\mathbf{I} - \tilde{\mathbf{G}}_c)^{k_1}, \quad (4)$$

where $\mathbf{D}_c$ is the degree matrix of $\mathbf{G}_c$, and $k_1$ is the order of the filter. Filter $\mathbf{F}_c$ has two important properties, one is that it can utilize information from neighbors within any order by adjusting the order $k_1$. When $k_1 = 1$, $\mathbf{F}_c$ is a linear filter that can merely utilize information from first-order neighbors of users in cluster $c$, however when $k_1 \geq 2$, $\mathbf{F}_c$ can recognize information from high-order neighbors of users. The other property is that it is a low-pass filter as shown in Proposition 3.1, which can preserve general information between users in that cluster, providing the big picture for user future interaction prediction [11].

PROPOSITION 3.1. *The filter $\mathbf{F}_c$ is a low-pass filter.*

$$\mathbf{G}_c = \mathbf{R}_c^T \mathbf{R}_c, \quad \tilde{\mathbf{G}}_c = \mathbf{D}_c^{-\frac{1}{2}} \mathbf{G}_c \mathbf{D}^{-\frac{1}{2}}, \quad (5)$$

$$\mathbf{F}_c = \mathbf{I} - (\mathbf{I} - \tilde{\mathbf{G}}_c)^{k_1}, \quad (6)$$

PROOF. It is obvious that $\tilde{\mathbf{G}}_c$ is a real and symmetric matrix, making the normalized Laplacian matrix $\tilde{\mathbf{L}}_c = \mathbf{I} - \tilde{\mathbf{G}}_c$ also a real and symmetric matrix. Thus $\tilde{\mathbf{L}}_c$ is diagonalizable and can be represented as $\tilde{\mathbf{L}}_c = \mathbf{U}_c \mathbf{\Lambda}_c \mathbf{U}_c^T$, and $\mathbf{U}_c \mathbf{U}_c^T = \mathbf{U}_c^T \mathbf{U}_c = \mathbf{I}$. Then we have

$$\begin{aligned} \mathbf{F}_c &= \mathbf{I} - (\mathbf{I} - \tilde{\mathbf{G}}_c)^{k_1} \\ &= \mathbf{U}_c \mathbf{U}_c^T - (\mathbf{U}_c \mathbf{\Lambda}_c \mathbf{U}_c^T)^{k_1} \\ &= \mathbf{U}_c \mathbf{U}_c^T - \mathbf{U}_c \mathbf{\Lambda}_c^{k_1} \mathbf{U}_c^T \\ &= \mathbf{U}_c (\mathbf{I} - \mathbf{\Lambda}_c^{k_1}) \mathbf{U}_c^T. \end{aligned} \quad (7)$$

We can observe that the frequency response function of $\mathbf{F}_c$ is $f(\lambda_{c,i}) = 1 - \lambda_{c,i}^{k_1}$, where $\lambda_{c,i}$ is the $i$-th eigenvalue of $\tilde{\mathbf{L}}_c$, and $\lambda_{c,i} \in [0, 1]$. By plotting the image of function $f(\lambda_{c,i})$ when $k_1 \geq 2$, we can find that the low frequency components are preserved whose function values are close to 1, and the high frequency components are removed whose function values are smaller than 1. Moreover, when $k_1$ becomes larger, more and more low frequency components will be preserved. Therefore, we can conclude that $\mathbf{F}_c$ is a low-pass filter. □

These two properties makes filter $\mathbf{F}_c$ able to extract rich and important information, so as to recognize user interaction patterns accurately. Then we can predict the future interactions $\mathbf{P}_c \in \mathbb{R}^{m_c \times n}$ of users in the cluster $c$

$$\mathbf{P}_c = \mathbf{R}_c \mathbf{F}_c. \quad (8)$$

The final prediction result of cluster-wise filter module is a concatenation of the prediction results of all clusters

$$\mathbf{P}^{(C)} = [\mathbf{P}_1^T || \cdots || \mathbf{P}_C^T]^T, \quad (9)$$

where || represents the concatenation operation. It is worth mentioning that the cluster-wise filter module in essence is a high-pass filter with localized low-pass characteristics, because within the cluster, it retains the general information from user interactions for user interaction pattern recognition, but between clusters, retaining the general information leads to significant differences in the user interaction patterns recognized from different clusters. Therefore,

it is a filter that exhibits low-pass characteristics at a local level but possesses high-pass characteristics at a global level.

Traditional collaborative filtering methods take the whole user interactions to recognize user interaction patterns, which will cause the recognized interaction patterns to be influenced by users who have different interaction behaviors, leading to sub-optimal user future interaction prediction. For instance, ItemCF [21] takes the whole user interactions to predict user future interactions

$$\mathbf{S}_{i,j} = \frac{\mathbf{R}_{:,i}^T \mathbf{R}_{:,j}}{||\mathbf{R}_{:,i}||_2 \cdot ||\mathbf{R}_{:,j}||_2}, \qquad \mathbf{P}^{(iCF)} = \mathbf{RS} \tag{10}$$

where $\mathbf{S}$ is the item similarity matrix, and we use cosine similarity method to calculate the similarity between items, and $\mathbf{R}_{:,i}$ is the interaction of the $i$-th item, i.e., the $i$-th column of interaction matrix $\mathbf{R}$. We can find that when predicting user $u$'s future interactions, all other users, whether similar or different in their interaction habits, are taken into consideration, making the recognized interaction patterns inaccurate. In Section 4.3, we compare the performance of ItemCF and Cluster-wise Filter Module with 1 layer to explore whether separating users is beneficial to user future interaction prediction, the results show that Cluster-wise Filter Module achieves better performance, which demonstrate the necessity of separating users with different interaction behaviors.

### 3.3 Globally-aware Filter Module

Cluster-wise filter module mainly focus on recognizing user unique interaction patterns, while user general interaction patterns recognized from all user interactions are also crucial for user preference modeling and future interaction prediction. Therefore, we propose a globally-aware filter module to recognize user general interaction patterns from interactions of all users. Existing methods [15, 22] usually adopt the ideal low-pass filter to recognize user general interaction patterns. However, the ideal low-pass filter in essence is equivalent to an infinite-layer spatial GCN [22], although it can utilize rich information from direct and distant neighbors for user general interaction patterns recognition, the over-smoothing issue seriously hurts the performance of the filter. In order to better recognize user general interaction patterns, we carefully devise the high-order low-pass filter $\mathbf{F}_H$ to be used together with the ideal low-pass filter $\mathbf{F}_I$ for user general interaction patterns recognition

$$\mathbf{F}_I = \mathbf{D}_I^{-\frac{1}{2}} \bar{\mathbf{U}} \bar{\mathbf{U}}^T \mathbf{D}_I^{\frac{1}{2}}, \quad \mathbf{F}_H = \mathbf{I} - (\mathbf{I} - \tilde{\mathbf{R}}^T \tilde{\mathbf{R}})^{k_2}, \tag{11}$$

where $\bar{\mathbf{U}}$ is the top-$K$ singular vectors of $\tilde{\mathbf{R}}$, and $k_2(k_2 \geq 2)$ is the order of $\mathbf{F}_H$. Compared to the ideal low-pass filter, the high-order low-pass filter limits the recognition of the user general interaction patterns to a specified neighborhood range, and as the order of the filter increases, the neighborhood range also increases, which alleviates the over-smoothing issue and makes the filter utilize information from nearby neighbors for user interaction pattern recognition. Combining these two filters can fully utilize the advantages of each filter, so that HiGSP can not only focus more on the information from the nearby neighborhood but also have access to information from the distant neighborhood. Therefore, the prediction of user future interaction through globally-aware filter module is

$$\mathbf{P}^{(G1)} = \mathbf{RF}_I, \quad \mathbf{P}^{(G2)} = \mathbf{RF}_H. \tag{12}$$

**Table 1: The detailed statistics of six real-world datasets.**

|  | # Users | # Items | # Interactions | Category |
|---|---|---|---|---|
| ML100K (MovieLens 100K) | 943 | 1,682 | 100,000 | Movie |
| Beauty (Amazon Beauty) | 22,363 | 12,101 | 198,502 | Product |
| BX (Book-Crossing) | 18,964 | 19,998 | 482,153 | Book |
| LastFM | 992 | 10,000 | 571,817 | Music |
| ML1M (MovieLens 1M) | 6,040 | 3,706 | 1,000,209 | Movie |
| Netflix | 20,000 | 17,720 | 5,678,654 | Movie |

### 3.4 Model Inference

Since cluster-wise filter module and globally-aware filter module recognize user interaction patterns from different perspectives, we believe combining them together can make them complement each other and make user preference modeling more accurate, thus we propose a linear model to combine the outputs of two modules as the final output of our proposed method

$$\mathbf{P} = \mathbf{P}^{(C)} + \alpha_1 \mathbf{P}^{(G1)} + \alpha_2 \mathbf{P}^{(G2)}, \tag{13}$$

where $\alpha_1$ and $\alpha_2$ are two weight coefficients.

### 3.5 The Time Complexity

The time complexity of Mixture-Of-Gaussian method for clustering is $O(MN^2)$, where we assume that the number of interactions and the number of components in algorithm are more less than the number of user/item. The time complexity of constructing filter in Eq.(6) is $O(k_1 \cdot N^3) \approx O(N^3)$, and that of predicting user future interactions is $O(MN^2)$. Therefore, the time complexity of the Cluster-wise Filter Module is $O(MN^2 + N^3)$. Similarly, the time complexity for the Globally-aware Filter Module is $O(N^3 + MN^2)$, where the time complexity of SVD is $O(N^3)$ according to GF-CF paper [22]. To sum up, suppose $M = \eta N$, the time complexity of our proposed method is $O(MN^2 + N^3) \approx O(N^3)$, which is equivalent to the time complexity of GF-CF and PGSP [15].

## 4 EXPERIMENT

We conduct extensive experiments to validate the effectiveness of HiGSP, so as to answer the following seven research questions:

- **RQ1:** How does HiGSP perform compared to other state-of-the-art GCN-based and GSP-based CF methods?
- **RQ2:** To what extent does each component affect the performance of HiGSP?
- **RQ3:** What's the impact of Cluster-wise Filter Module and Globally-aware Filter Module on the user preference modeling?
- **RQ4:** Is separating users with different interaction behaviors necessary when recognizing user interaction patterns?
- **RQ5:** Is high-order low-pass filter more effective than ideal low-pass filter when predicting user future interactions?
- **RQ6:** Is HiGSP highly efficient compared to other state-of-the-art CF methods?
- **RQ7:** How will the number of clusters, an important hyper-parameter, impact the performance of HiGSP?

### 4.1 Experimental Settings

We conduct experiments on six real-world datasets to verify the efficacy of HiGSP: **ML100K**, **ML1M**, **Netflix**, **Beauty**, **BX**, and

**Table 2: The performance comparison on six public datasets. The best performance is denoted in bold, the second best performance is denoted with an underline. DGCF occurred the OOM problem on Netflix, so we do not report the results.**

| | | LR-GCCF | LCFN | DGCF | LightGCN | IMP-GCN | SimpleX | UltraGCN | GF-CF | PGSP | HiGSP |
|---|---|---|---|---|---|---|---|---|---|---|---|
| ML100K | F1@5 | 0.1048 | 0.0972 | 0.1632 | 0.1714 | 0.1588 | 0.1683 | 0.1613 | 0.1622 | 0.1667 | **0.1732** |
| | MRR@5 | 0.4943 | 0.4852 | 0.6369 | 0.6362 | 0.6006 | 0.6397 | 0.6094 | 0.6234 | 0.6149 | **0.6629** |
| | NDCG@5 | 0.5587 | 0.5542 | 0.6998 | 0.7034 | 0.6764 | 0.6995 | 0.6786 | 0.6875 | 0.6851 | **0.7166** |
| | F1@10 | 0.1444 | 0.1393 | 0.2421 | 0.2461 | 0.2287 | 0.2466 | 0.2366 | 0.2425 | 0.2407 | **0.2522** |
| | MRR@10 | 0.4616 | 0.4142 | 0.6012 | 0.5877 | 0.5690 | 0.6064 | 0.5749 | 0.6010 | 0.5767 | **0.6384** |
| | NDCG@10 | 0.5603 | 0.5326 | 0.6808 | 0.6771 | 0.6605 | 0.6866 | 0.6688 | 0.6843 | 0.6722 | **0.7021** |
| ML1M | F1@5 | 0.0618 | 0.0577 | 0.1378 | 0.1452 | 0.1309 | 0.1490 | 0.1378 | 0.1520 | 0.1510 | **0.1573** |
| | MRR@5 | 0.2997 | 0.2902 | 0.5064 | 0.5160 | 0.4934 | 0.5244 | 0.5087 | 0.5254 | 0.5313 | **0.5422** |
| | NDCG@5 | 0.3540 | 0.3421 | 0.5709 | 0.5845 | 0.5583 | 0.5934 | 0.5739 | 0.5935 | 0.5963 | **0.6072** |
| | F1@10 | 0.0930 | 0.0860 | 0.1956 | 0.2044 | 0.1837 | 0.2087 | 0.1963 | 0.2106 | 0.2090 | **0.2196** |
| | MRR@10 | 0.2946 | 0.2839 | 0.4825 | 0.5010 | 0.4685 | 0.5051 | 0.4886 | 0.4996 | 0.5063 | **0.5156** |
| | NDCG@10 | 0.3787 | 0.3694 | 0.5740 | 0.5873 | 0.5594 | 0.5921 | 0.5773 | 0.5897 | 0.5923 | **0.6042** |
| Beauty | F1@5 | 0.0285 | 0.0100 | 0.0357 | 0.0359 | 0.0309 | 0.0365 | 0.0324 | 0.0362 | 0.0368 | **0.0399** |
| | MRR@5 | 0.0423 | 0.0147 | 0.0523 | 0.0530 | 0.0444 | 0.0544 | 0.0476 | 0.0523 | 0.0538 | **0.0578** |
| | NDCG@5 | 0.0533 | 0.0191 | 0.0663 | 0.0668 | 0.0570 | 0.0682 | 0.0599 | 0.0659 | 0.0674 | **0.0723** |
| | F1@10 | 0.0268 | 0.0094 | 0.0325 | 0.0327 | 0.0289 | 0.0333 | 0.0287 | 0.0330 | 0.0335 | **0.0353** |
| | MRR@10 | 0.0443 | 0.0147 | 0.0507 | 0.0519 | 0.0451 | 0.0517 | 0.0441 | 0.0517 | 0.0525 | **0.0537** |
| | NDCG@10 | 0.0635 | 0.0231 | 0.0750 | 0.0763 | 0.0667 | 0.0761 | 0.0649 | 0.0751 | 0.0765 | **0.0778** |
| LastFM | F1@5 | 0.0405 | 0.0293 | 0.0487 | 0.0559 | 0.0457 | 0.0542 | 0.0463 | 0.0545 | 0.0539 | **0.0592** |
| | MRR@5 | 0.5160 | 0.4378 | 0.5769 | 0.6306 | 0.5535 | 0.6104 | 0.5666 | 0.6343 | 0.6200 | **0.6397** |
| | NDCG@5 | 0.5789 | 0.4984 | 0.6408 | 0.6876 | 0.6206 | 0.6701 | 0.6269 | 0.6854 | 0.6764 | **0.6962** |
| | F1@10 | 0.0689 | 0.0507 | 0.0867 | 0.0968 | 0.0799 | 0.0941 | 0.0811 | 0.0964 | 0.0945 | **0.1003** |
| | MRR@10 | 0.5046 | 0.4280 | 0.5706 | 0.6073 | 0.5398 | 0.5930 | 0.5529 | 0.6086 | 0.6007 | **0.6135** |
| | NDCG@10 | 0.5893 | 0.5171 | 0.6509 | 0.6855 | 0.6241 | 0.6690 | 0.6340 | 0.6817 | 0.6767 | **0.6861** |
| BX | F1@5 | 0.0138 | 0.0144 | 0.0311 | 0.0333 | 0.0142 | 0.0373 | 0.0319 | 0.0288 | 0.0288 | **0.0376** |
| | MRR@5 | 0.0284 | 0.0255 | 0.0580 | 0.0620 | 0.0319 | 0.0695 | 0.0602 | 0.0566 | 0.0555 | **0.0744** |
| | NDCG@5 | 0.0364 | 0.0344 | 0.0725 | 0.0766 | 0.0402 | 0.0864 | 0.0743 | 0.0704 | 0.0691 | **0.0912** |
| | F1@10 | 0.0142 | 0.0140 | 0.0319 | 0.0334 | 0.0155 | **0.0390** | 0.0319 | 0.0309 | 0.0308 | 0.0385 |
| | MRR@10 | 0.0274 | 0.0276 | 0.0580 | 0.0601 | 0.0331 | 0.0696 | 0.0594 | 0.0576 | 0.0578 | **0.0710** |
| | NDCG@10 | 0.0414 | 0.0411 | 0.0839 | 0.0861 | 0.0479 | 0.0984 | 0.0847 | 0.0821 | 0.0817 | **0.0988** |
| Netflix | F1@5 | 0.0546 | 0.0394 | – | 0.0709 | 0.0611 | 0.0658 | 0.0503 | 0.0739 | 0.0708 | **0.0764** |
| | MRR@5 | 0.5323 | 0.3982 | – | 0.6147 | 0.5528 | 0.5902 | 0.4941 | 0.6360 | 0.6127 | **0.6532** |
| | NDCG@5 | 0.6023 | 0.4726 | – | 0.6822 | 0.6328 | 0.6603 | 0.5680 | 0.7005 | 0.6811 | **0.7162** |
| | F1@10 | 0.0920 | 0.0660 | – | 0.1205 | 0.1026 | 0.1137 | 0.0835 | 0.1244 | 0.1185 | **0.1285** |
| | MRR@10 | 0.5234 | 0.4056 | – | 0.5974 | 0.5386 | 0.5839 | 0.4708 | 0.6126 | 0.5905 | **0.6275** |
| | NDCG@10 | 0.6134 | 0.5058 | – | 0.6814 | 0.6307 | 0.6694 | 0.5746 | 0.6928 | 0.6756 | **0.7062** |

**LastFM**. All datasets are divided into training set and test set with the ratio of 8:2, and we take the 10% training set as validation set for hyper-parameter tuning. Table 1 shows the statistics of six datasets.

We compare the performance of HiGSP with various state-of-the-art baselines, including seven GCN- based CF methods and two GSP-based CF methods: (1) **LR-GCCF** [4]; (2) **LCFN** [35]; (3) **DGCF** [26]; (4) **LightGCN** [10]; (5) **IMP-GCN** [14]; (6) **SimpleX** [16]; (7) **UltraGCN** [17]; (8) **GF-CF** [22]; (9) **PGSP** [15].

We evaluate the performance of HiGSP with three popular used metrics in the Top-K recommendation scenario: (1) **F1**; (2) **Mean Reciprocal Rank (MRR)**; and (3) **Normalized Discounted Cumulative Gain (NDCG)**. For each metric, we report their results when K=5 and K=10 to comprehensively evaluate the performance of HiGSP.

For all baselines, we use their released code and carefully tune hyper-parameters. For HiGSP, we tune the number of clusters $C$ from 2 to 30, orders $k_1$ and $k_2$ from 2 to 12, primary components $Q$ from 32 to 1024, and coefficients $\alpha_1$ and $\alpha_2$ from 0.1 to 1.0. Due to the space limitation, we leave the details about experimental settings in the Appendix.

## 4.2 RQ1: Performance Comparison

Table 2 shows the performance comparison results of all methods, and we have the following observations:

1) Among all GCN-based CF methods, LightGCN and SimpleX have achieved a leading position in most metrics. Because LightGCN removes the two inefficacy components in traditional GCN, i.e., feature transformation and non-linear activation, to improve the accuracy of future interaction prediction. While SimpleX designs an appropriate loss function with negative sampling strategy, thereby improving the accuracy of interaction prediction.

2) GF-CF and PGSP are comparable to or even better than Light-GCN and SimpleX in most cases. This is because the ideal low-pass

filter they adopt is equivalent to an infinite-layer spatial GCN, which can obtain richer information from direct and distant neighbors compared to other GCN-based methods which can merely obtain information from nearby neighbors, thereby improving the accuracy of user preference modeling.

3) HiGSP achieves promising results in all metrics, significantly outperforming other methods. We attribute this to the cluster-wise filter module and globally-aware filter module that recognize user-matched interaction patterns, thereby modeling user preference and predicting user future interaction prediction accurately.

### 4.3 RQ2: Ablation Study

We conduct the ablation study on BX, ML1M and Netflix to analyze the impacts of cluster-wise filter module(CwFilter), ideal low-pass filter and high-order low-pass filter in globally-aware filter module (GaFilter-i and GaFilter-h), and different clustering methods (K-means/KMS, Agglomerative Clustering/AGG and Birch/BIRCH) on the performance of HiGSP. We also analyse the performance of ItemCF and CwFilter. Table 7 shows the results. Due to the space limitation, we only report the results at $K$=5, and the results at $K$=10 are presented in the Appendix. From the results, we have the following findings:

1) Comparing settings (a) and (b), we find that the performance of HiGSP decreases when removing cluster-wise filter module. This is because this module can recognize user unique interaction patterns that reflect user preference without being influenced by other unrelated users from user interactions, which is beneficial to user preference modeling, thus predicting accurate interactions.

2) Comparing settings (a) and (e), we find that globally-aware filter module contributes to user future interaction prediction, since it can recognize user general interaction patterns to improve the accuracy of user preference modeling. We further observe that separately removing ideal low-pass filter and high-order filter from globally-aware filter module reduces the model performance by comparing setting (c)–(e), since the user general interaction patterns recognized by these two filters are complementary, and combining them can make user preference modeling more accurate.

3) Comparing settings (a) and (f)–(h), we find that Mixture-Of-Gaussian method (setting (a)) achieves better performance in most cases, which indicates that compared to AGG, BIRCH and KMS, the Mixture-Of-Gaussian method can more accurately recognize the similarity of user interaction patterns, thereby accurately clustering users based on their preference.

4) Comparing settings (i) and (j), we can find that CwFilter achieves better results than ItemCF, which demonstrates the necessity of separating users with different interaction behaviors into different clusters when predicting user future interactions, making the recognized interaction patterns more accurate, thereby obtaining more accurate interaction prediction.

### 4.4 RQ3: Case study

We study the impacts of cluster-wise filter module and globally-aware filter module on the performance of HiGSP by analyzing the matching degree of user historical preference distribution (calculated from training data) and user predicted preference distribution (calculated from model prediction) on ML100K. Specifically, we

define user $u$'s historical preference distribution $p_u$ according to the categories (e.g., Comedy) of items that user has interacted with:

$$p_u(\text{category} = l) = C_{ul}/\textstyle\sum_{k=1}^{L} C_{uk}, \quad (14)$$

where $C_{ul}$ ($u = 1, \cdots, M, \; l = 1, \cdots, L$) is the number of appearances of the $l$-th category in user $u$'s interacted items, $M$ is the number of users, and $L$ is the number of categories.

Similarly, we can define user $u$'s predicted preference distributions from his/her predicted items in test using CwFilter, GaFilter and CwFilter+GaFilter (HiGSP) as $q_u^{(1)}$, $q_u^{(2)}$ and $q_u^{(3)}$ respectively. Then we use the Kullback–Leibler (KL) divergence to evaluate the matching degree of the historical preference distribution and the predicted preference distribution, where the smaller the value of KL divergence, the more consistent historical preference and predicted preference are. The KL divergence between $p$ and $q$ is:

$$\mathbf{KL}(p, q^{(w)}) = \frac{1}{M} \sum_{u,l} p_u(l) \ln \frac{p_u(l)}{q_u^{(w)}(l)}, \; w = 1, 2, 3. \quad (15)$$

Figure 4 shows the results of KL divergence with respect to $K(K \in [8, 18])$ most popular categories, we find that the KL divergence of CwFilter+GaFilter is smaller than that of CwFilter and GaFilter in all setting of $K$, which implies both cluster-wise filter module and globally-aware filter module are beneficial to user preference modeling. The former can recognize user unique interaction patterns and the latter can recognize user general interaction patterns, thereby making the user preference modeling more accurate. It is noted that the KL divergence increases when $K$ becomes larger, it is because a movie may be associated with multiple categories and a user's viewing behavior may not necessarily reflect an interest in all of those categories, several categories at the tail are considered as noisy, and taking them into account will affect the analysis of preference matching degree.

We take user 368 as an example to show that with CwFilter and GaFilter, HiGSP can recommend items more suitable for user preference, indicating the effectiveness of CwFilter and GaFilter. Figure 5 shows the user historical preference distribution of user 368 with respect to Top-8 preferred categories, and Table 4—6 show the categories of Top-5 recommended items predicted by CwFilter, GaFilter and HiGSP respectively. Comparing three tables, we can find that HiGSP recommends more items related to user's favorite category, i.e., Drama, and places those items at the top of the recommendation list compared to CwFilter and GaFilter. Moreover,

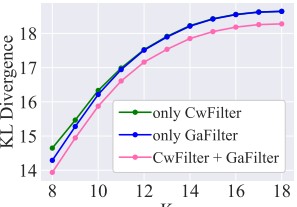

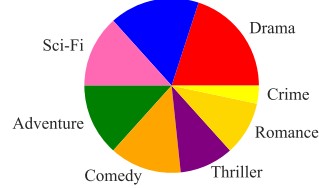

**Figure 4: The user preference matching degree of HiGSP under different modules on ML100K dataset.**

**Figure 5: The user historical preference distribution of user 368 with respect to Top-8 his/her preferred categories.**

**Table 3: The ablation study of HiGSP on BX, ML1M and Netflix datasets.**

| | BX | | | ML1M | | | Netflix | | |
|---|---|---|---|---|---|---|---|---|---|
| | F1@5 | MRR@5 | NDCG@5 | F1@5 | MRR@5 | NDCG@5 | F1@5 | MRR@5 | NDCG@5 |
| (a) HiGSP | 0.0376 | 0.0744 | 0.0912 | 0.1573 | 0.5422 | 0.6072 | 0.0764 | 0.6532 | 0.7162 |
| (b) HiGSP w/o CwFilter | 0.0291 | 0.0578 | 0.0714 | 0.1561 | 0.5374 | 0.6052 | 0.0756 | 0.6490 | 0.7126 |
| (c) HiGSP w/o GaFilter-i | 0.0371 | 0.0740 | 0.0900 | 0.1497 | 0.5263 | 0.5943 | 0.0764 | 0.6526 | 0.7162 |
| (d) HiGSP w/o GaFilter-h | 0.0336 | 0.0673 | 0.0828 | 0.1539 | 0.5327 | 0.5991 | 0.0759 | 0.6493 | 0.7131 |
| (e) HiGSP w/o GaFilter-i+GaFilter-h | 0.0324 | 0.0649 | 0.0797 | 0.1439 | 0.5128 | 0.5807 | 0.0753 | 0.6456 | 0.7105 |
| (f) HiGSP w/ KMS | 0.0324 | 0.0630 | 0.0781 | 0.1573 | 0.5421 | 0.6070 | 0.0765 | 0.6539 | 0.7173 |
| (g) HiGSP w/ AGG | 0.0311 | 0.0616 | 0.0765 | 0.1568 | 0.5411 | 0.6073 | 0.0761 | 0.6508 | 0.7146 |
| (h) HiGSP w/ BIRCH | 0.0375 | 0.0728 | 0.0893 | 0.1549 | 0.5375 | 0.6037 | 0.0762 | 0.6505 | 0.7143 |
| (i) Item CF | 0.0275 | 0.0568 | 0.0692 | 0.1190 | 0.4544 | 0.5197 | 0.0552 | 0.5265 | 0.6009 |
| (j) CwFilter (1 Layer, i.e., $k_1 = 1$) | 0.0331 | 0.0639 | 0.0791 | 0.1282 | 0.4750 | 0.5412 | 0.0600 | 0.5568 | 0.6260 |

**Table 4: The categories of Top-5 recommended items predicted by CwFilter.**

| ItemID | Categories | | | | |
|---|---|---|---|---|---|
| 474 | Drama | | | | |
| 257 | Drama | Sci-Fi | | | |
| 170 | Comedy | Sci-Fi | | | |
| 895 | Drama | | | | |
| 180 | Action | Adventure | Romance | Sci-Fi | War |

**Table 5: The categories of Top-5 recommended items predicted by GaFilter.**

| ItemID | Categories | | |
|---|---|---|---|
| 314 | Drama | Thriller | |
| 749 | Drama | | |
| 894 | Horror | Thriller | |
| 312 | Action | Drama | Romance |
| 895 | Drama | | |

**Table 6: The categories of Top-5 recommended items predicted by HiGSP.**

| ItemID | Categories | | |
|---|---|---|---|
| 314 | Drama | Thriller | |
| 749 | Drama | | |
| 895 | Drama | | |
| 312 | Action | Drama | Romance |
| 342 | Action | Horror | Sci-Fi |

HiGSP also recommends more items related to Action, which is the second preferred category for user 368. Therefore, combining CwFilter and GaFilter can model user preference more accurately.

### 4.5 RQ4: Visualization

We analyze the necessity of separating users with different interaction behaviors when recognizing user interaction patterns by plotting the heat maps of the filters in the cluster-wise filter module on ML100K . Figure 6 shows the heat maps of the filters of 3 randomly selected clusters (out of total 25 clusters), and we leave the heat maps of other filters in the Appendix due to the space limitation. As a comparison, Figure 7 shows the heat map of the filter corresponding to the cluster that contains all users, i.e., no clustering in the cluster-wise filter module. Note that we only show the relationship between item 1—30 in the filter for better presentation.

From the results, it is obvious that the relationships between items change across different clusters, and each relationship is significantly different from that in the case of no clustering, which indicates that different clusters contains different user interaction patterns. Compared to mixing them to model user preference, separately modeling user preference is better since each interaction pattern can be easily recognized and will not influence others, thereby predicting user future interactions accurately.

### 4.6 RQ5: High-order Low-pass Filter Analysis

As describe in Section 3.3, ideal low-pass filter can extract information from infinite-order neighbors since it is equal to an infinite-layer GCN, while over-smoothing issue limits its performance. High-order low-pass filter focuses on the information from nearby neighbors, thus will not be easily influenced by the over-smoothing

issue, and it can extract information from any order neighbors by adjusting the layer number/order $k_2$, thus the latter is more effective than the former when modeling user preference and predicting future interactions. To demonstrate this claim, we compare the performance of GaFilter-i and GaFilter-h on ML100K and ML1M. Figure 8 shows the results of ML1M, and we leave the results of ML100K in the Appendix due to the space limitation.

From the results, we can find that when the number of layer of GaFilter-h increases, the performance also increases and exceeds GaFilter-i when the number of layer is larger than 6. This is because more and more information from neighbors can be used to model user preference, making future interaction prediction more accurate. However, when the number of layers continues to increase, the performance of GaFilter-h decreases because it is also affected by the over-smoothing issue. In our empirical study, the performance of GaFilter-h and GaFilter-i is roughly equivalent when layer number is 60. In practice, we need to balance the performance of GaFilter-h and the time cost of building GaFilter-h, thus we will choose an appropriate value for layer number, such as 10, 15. In this case, GaFilter-h is more efficient than GaFilter-i when modeling user preference and predicting user future interactions.

### 4.7 RQ6: Efficiency Analysis

We conduct the efficiency analysis on ML1M by comparing the training time of HiGSP and other CF methods. For LightGCN, SimpleX and UltraGCN which need back propagation to train the model, we accumulate the training time until we obtain the optimal validation results. For GF-CF, PGSP and HiGSP, we directly calculate its training time, including time for SVD. Figure 9 shows the results, we can find that HiGSP is comparable to GSP-based methods and

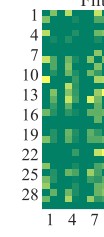
Filter of 3-th cluster

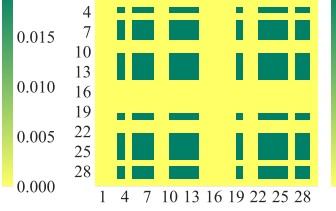
Filter of 12-th cluster

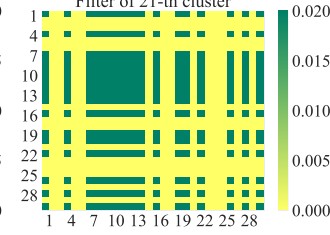
Filter of 21-th cluster

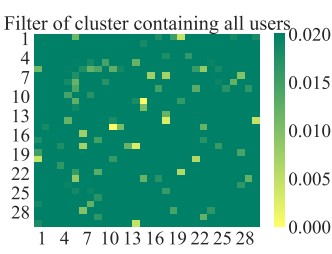
Filter of cluster containing all users

**Figure 6: The heat maps of the filters of 3 randomly selected clusters (out of total 25 clusters) in the cluster-wise filter module on ML100K.**

**Figure 7: The heat map of filter of cluster containing all users.**

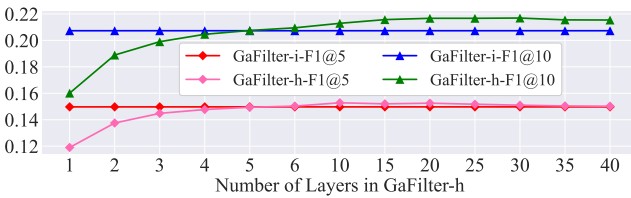

**Figure 8: The performance comparison between GaFilter-i and GaFilter-h on ML1M.**

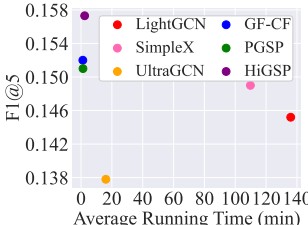
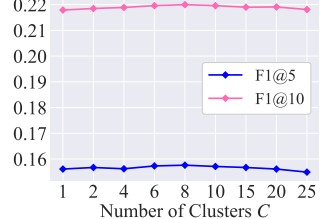

**Figure 9: The average running time (5 times) vs. F1@5 of HiGSP and other state-of-the-art methods on ML1M.**

**Figure 10: The sensitivity results of the number of clusters in cluster-wise filter module on ML1M dataset.**

more efficient than the GCN-based methods, while achieving the best performance. Therefore, we can conclude that HiGSP is highly efficient and highly effective.

## 4.8 RQ7: Sensitivity Analysis

We analyze the impact of the number of clusters on the performance of HiGSP when it varies from 1 to 25 on ML100K dataset. Figure 10 shows the results of F1@5 and F1@10 for better presentation. From the results, we can find that the performance of HiGSP first increases and then decreases, it is because separately recognizing user interaction patterns can prevent them from influencing each other, thereby making user preference modeling more accurately. However, too many clusters will further divided users with similar preference into different clusters, result in insufficient user preference modeling and poor model performance. Note that the performance of HiGSP when $C \geq 2$ is better than that when $C = 1$, indicating the necessity of cluster-wise filter module.

## 5 RELATED WORK

### 5.1 GCN-based Recommendation

Nowadays, graph convolutional networks (GCNs) [12, 29] are widely used in the realm of recommendation algorithms [8, 13, 24, 26–28, 30, 32, 34]. By leveraging the powerful structural feature extraction ability of GCN [33, 36], rich user preference information can be extracted from a bipartite graph composed of user interactions. GC-MC [2] proposed an auto-encoder framework that combined GCN with matrix completion to predict missing values in user-item interaction matrices. NGCF [25] proposed a GCN-based recommendation framework that explicitly encodes the collaborative signal in the form of high-order connectivities by performing embedding propagation. LightGCN [10] removed two common designs in GCN, i.e., feature transformation and nonlinear activation, and proposed a lightweight framework that only preserved the most essential component in GCN—neighborhood aggregation—for GCN-based collaborative filtering. IMPGCN [14] proposed to extract features from sub-graphs which consist of users with similar interests and their interacted items to make recommendation.

### 5.2 GSP-based Recommendation

GSP-based recommendation methods have attracted more and more researchers' attention due to its excellent prediction performance and high training and inference efficiency [11, 15, 22]. GF-CF [22] developed a unified graph convolution-based framework and explored the connections between the collaborative filtering methods (e.g., neighborhood-based methods [1], low-rank matrix factorization [3]) and the low-pass filters, and proposed a simple yet effective collaborative filtering method that equipped with a linear filter and an ideal low-pass filter to model user preference and make recommendation. PGSP [15] proposed a mixed-frequency low-pass filter over the personalized graph signal to model user preference and predict user interactions.

## 6 CONCLUSION

We propose a hierarchical graph signal processing method (HiGSP) for collaborative filtering, which consists of a cluster-wise filter module and a globally-aware filter module to recognize user unique and general interaction patterns respectively for user preference modeling. Extensive experiments demonstrate the superiority of HiGSP compared to other GCN-based and GSP-based recommendation methods in terms of efficacy and efficiency.

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

# A DETAILS OF EXPERIMENTAL SETTINGS

## A.1 Experimental Settings

*A.1.1 Datasets.* We conduct experiments on six real-world datasets from four domains to verify the efficacy of HiGSP: (1) **ML100K**, **ML1M**, **Netflix** (three movie datasets); (2) **Beauty** (a product dataset); (3)**BX** (a book dataset), and (4)**LastFM** (a music dataset). All datasets are divided into training set and test set with the ratio of 8:2, and we take the 10% training set as validation set for hyper-parameter tuning.

*A.1.2 Baselines.* We compare the performance of HiGSP with various state-of-the-art baselines, including seven GCN- based CF methods and two GSP-based CF methods: (1) **LR-GCCF** [4] is a GCN-based CF method that removes non-linear activation and introduces the skip connection [9] to ease the model training and alleviate the over-smoothing problem. (2) **LCFN** [35] is a GCN-based method that leverages the original graph convolution in GCN [12] and proposes a Low-pass Collaborative Filter to remove the noise in the data. (3) **DGCF** [26] is a GCN-based CF method that models a distribution over intents for each user-item interaction to iteratively refine the intent-aware interaction graphs, and encourages independence of different intents to yield disentangled representations. (4) **LightGCN** [10] is a simple yet effective GCN-based CF method that removes feature transformation and non-linear activation to improve both efficiency and accuracy. (5) **IMP-GCN** [14] is a GCN based method that propagates information in the user sub-graphs to reduce the impact of noise or negative information and alleviate the over-smoothing problem.(6) **SimpleX** [16] is a GCN-based CF

**Table 7: The ablation study of HiGSP on BX, ML1M and Netflix datasets.**

| | BX | | | ML1M | | | Netflix | | |
|---|---|---|---|---|---|---|---|---|---|
| | F1@10 | MRR@10 | NDCG@10 | F1@10 | MRR@10 | NDCG@10 | F1@10 | MRR@10 | NDCG@10 |
| (1) HiGSP | 0.0385 | 0.0710 | 0.0988 | 0.2196 | 0.5156 | 0.6042 | 0.1285 | 0.6275 | 0.7062 |
| (2) HiGSP w/o CwFilter | 0.0305 | 0.0575 | 0.0815 | 0.2169 | 0.5112 | 0.5999 | 0.1275 | 0.6245 | 0.7035 |
| (3) HiGSP w/o GaFilter-i | 0.0366 | 0.0676 | 0.0936 | 0.2130 | 0.5092 | 0.5979 | 0.1281 | 0.6260 | 0.7051 |
| (4) HiGSP w/o GaFilter-h | 0.0353 | 0.0661 | 0.0926 | 0.2162 | 0.5108 | 0.5990 | 0.1278 | 0.6256 | 0.7044 |
| (5) HiGSP w/o GaFilter-i+GaFilter-h | 0.0335 | 0.0615 | 0.0863 | 0.2045 | 0.4954 | 0.5853 | 0.1266 | 0.6204 | 0.7012 |
| (6) HiGSP w/o KMS | 0.0324 | 0.0576 | 0.0824 | 0.2196 | 0.5154 | 0.6041 | 0.1285 | 0.6277 | 0.7065 |
| (7) HiGSP w/o AGG | 0.0311 | 0.0565 | 0.0801 | 0.2191 | 0.5125 | 0.6023 | 0.1284 | 0.6251 | 0.7053 |
| (8) HiGSP w/o BIRCH | 0.0373 | 0.0666 | 0.0939 | 0.2187 | 0.5187 | 0.6056 | 0.1280 | 0.6244 | 0.7047 |
| (9) ItemCF | 0.0298 | 0.0545 | 0.0763 | 0.1601 | 0.4126 | 0.5111 | 0.0916 | 0.5187 | 0.6132 |
| (10) CwFilter (1 Layer, i.e., $k_1 = 1$) | 0.0335 | 0.0610 | 0.0871 | 0.1771 | 0.4444 | 0.5392 | 0.0993 | 0.5323 | 0.6261 |

method that focuses on the choice of loss function and negative sampling ratio, forming a simple but strong baseline with the proposed cosine contrastive loss and large negative sampling ratio. (7) **UltraGCN** [17] is a GCN-based CF method which can approximate the limit of infinite-layer graph convolutions via a constraint loss and allows for more appropriate edge weight assignments and flexible adjustment of the relative importance among different types of relationships. (8) **GF-CF** [22] is a simple yet effective GSP-based CF method that integrates a linear filter and an ideal low-pass filter to model user preference. Note that GF-CF is a parameter-free method and does not suffer from time-consuming training phase. (9) **PGSP** [15] is a GSP-based CF method that uses a mixed-frequency low-pass filter over the personalized graph signal to predict user future interactions. PGSP is also a parameter-free method and does not suffer from time-consuming training phase.

*A.1.3 Metrics.* We evaluate the performance of HiGSP with three popular used metrics in the Top-K recommendation scenario: (1) **F1**, which balances between precision and recall by harmonic mean; (2) **Mean Reciprocal Rank (MRR)**, which evaluates the performance of ranking according to the harmonic mean of the ranks; and (3) **Normalized Discounted Cumulative Gain (NDCG)**, which accumulate the gains from ranking list with the discounted gains at lower ranks. For each metric, we report their results when K=5 and K=10 to comprehensively evaluate the performance of HiGSP.

*A.1.4 Hyper-parameter Settings.* For all baselines, we use their released code and carefully tune hyper-parameters. For HiGSP, we tune the number of clusters $C$ from 2 to 30, orders $k_1$ and $k_2$ from 2 to 12, primary components $Q$ from 32 to 1024, and coefficients $\alpha_1$ and $\alpha_2$ from 0.1 to 1.0.

## B  ADDITIONAL RESULTS ABOUT ABLATION STUDY

Table 7 shows the additional results of ablation study with respect to the metrics at $K = 10$ on BX, ML1M and Netflix datasets, we can find that both cluster-wise filter module and global-aware filter module are beneficial to user preference modeling and future interaction prediction, and HiGSP achieves better results when equipped with

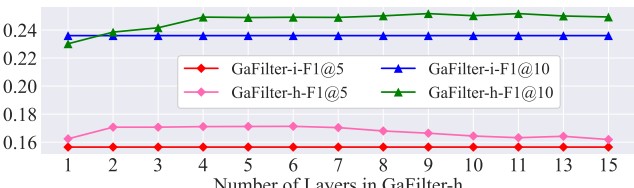

**Figure 11: The performance comparison between GaFilter-i and GaFilter-h on ML100K.**

Mix-of-Gaussian method compare with other clustering methods. Moreover, separating users into different clusters is necessary.

## C  ADDITIONAL RESULTS ABOUT VISUALIZATION

Figure 12 shows the results of all 25 heat maps of filters, and as a comparison, Figure 7 shows the heat map of the filter corresponding to the cluster that contains all users, i.e., no clustering in cluster-wise filter module. It is noted that to ease of the presentation, we only show the relationship between item 1–30 in the filter.

From the results, it is obvious that the relationships between items change with different clusters, and each relationship is different from that in the case of no clustering, which indicates that different clusters contains different user preference, which can be seen as the interactive transfer measurement between items. Compared to mixing them together when modeling user preference, separately modeling user preference is better since user unique interaction patterns can be easily recognized, thereby improving the accuracy of future interaction prediction.

## D  ADDITIONAL RESULTS ABOUT HIGH-ORDER LOW-PASS FILTER ANALYSIS

Figure 11 shows the additional results of ML100K. From the results, we can find that when the number of layer of GaFilter-h increase, the performance also increases and exceeds GaFilter-i when the number of layers is larger than 2. This is because more and more information from neighbors can be used to model user preference, making interaction prediction more accurate. However, when the number

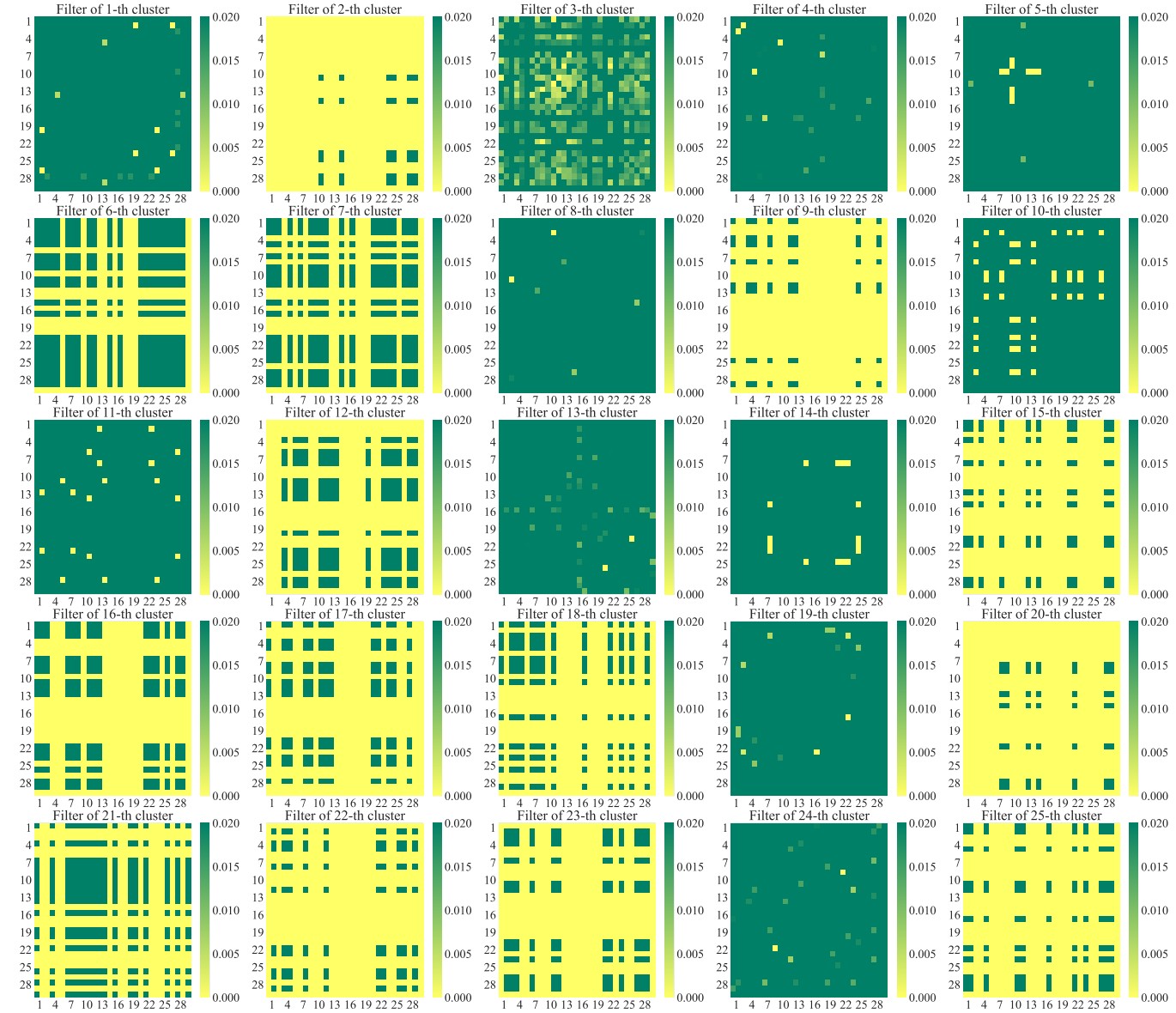

**Figure 12: The heat maps of 25 clusters in the cluster-wise filter module on ML100K dataset.**

of layers continues to increase, the performance of GaFilter-h decreases because it is also affected by the over-smoothing problem. In our empirical study, the performance of GaFilter-h and GaFilter-i is roughly equivalent when layer number is 35. In practice, we need to balance the performance of GaFilter-h and the time cost of building GaFilter-h, thus we will choose an appropriate value for layer number, such as 8, 10. In this case, GaFilter-h is more efficient than GaFilter-i when predicting user future interactions.

