# OpenReview forum: "Hierarchical Graph Signal Processing for Collaborative Filtering"
_ACM.org/TheWebConf/2024/Conference — TheWebConf24_

### Official Review · Reviewer_fYXW · 2023-11-24

**Novelty:** 5
**Technical Quality:** 5

**Review:**

The manuscript proposes a hierarchical graph signal processing (HiGSP) method for collaborative filtering in recommender systems. The method comprises two key modules: a cluster-wise filter module that identifies unique user interaction patterns from interactions of users with similar preferences, and a globally-aware filter module that identifies general user interaction patterns from all user interactions. The manuscript conducts extensive experiments on several datasets and demonstrates the superiority of HiGSP compared to other recommendation methods.

### Pros:
1. The proposed hierarchical graph signal processing (HiGSP) method is innovative and addresses the issue of user interaction pattern recognition in recommender systems.
2. The manuscript provides a well-structured review of relevant literature, discussing the limitations of existing methods.
3. The manuscript conducts extensive experiments on six real-world datasets, demonstrating the superiority of HiGSP compared to other state-of-the-art methods.

### Cons:
1. The motivation for this manuscript is not very clear. Although the authors provided an intuitive explanation in Figure 1, they did not clearly articulate the significant differences from mainstream cluster-based item CF methods.
2. The proposed method section of this manuscript lacks sufficient logic, and the model structure diagram in Figure 3 is not consistent with the manuscript's description.
3. In the introduction, it is mentioned that the clustering module focuses on capturing unique user interaction patterns, while in the method section at line 283, it is suggested that it can retain general information, which is contradictory.
4. The method section lacks emphasis on key details, such as the loss function used and the training mode (end-to-end training or two-stage?).

**Questions:**

1. How is "hierarchical" reflected in this manuscript? Is a hierarchical method designed, or is a hierarchical representation obtained? This is not well demonstrated in the current version.
2. Given that pattern recognition is based on clusters of similar users, is there a risk of losing diversity? As we know, collaborative filtering can discover potentially diverse items that users might be interested in based on their behavioral information. Therefore, does the clustering method restrict user interests and potentially compromise the essence of collaborative filtering?
3. How is efficiency reflected? Can more details about the runtime in Figure 9 be provided? We can see that the time difference between the proposed method and PGSP is not significant. Was the time of the clustering algorithm considered during the experiments?
4. There is a noteworthy point for discussion regarding whether it is better to use behavioral information or user attribute information when calculating user clusters. Since this manuscript focuses on behavioral information, is there potential overlap between the clustering-wise filter stage and the globally-aware filter stage?

**Reviewer Confidence:**

4: The reviewer is certain that the evaluation is correct and very familiar with the relevant literature

**Scope:**

4: The work is relevant to the Web and to the track, and is of broad interest to the community

---

### Official Review · Reviewer_RLBg · 2023-11-24

**Novelty:** 5
**Technical Quality:** 5

**Review:**

This paper presents a a hierarchical graph signal processing method, HiGSP, for collaborative filtering. The authors attempt to segment users into distinct groups based on shared characteristics, cluster users based on their interactions, and use a cluster-wise filter mechanism to recognize the interaction patterns of users in their clusters. Experiments on six real-world datasets verify the effectiveness and efficiency superiority of HiGSP.

In terms of quality, the proposed HiGSP model for collaborative filtering is perceived as technically sound. The empirical study is comprehensive, and the experiment results convincingly demonstrate the superiority of HiGSP in both effectiveness and efficiency for the top-k recommendation task.

Regarding clarity, there is room for improvement in the paper's presentation. Section 2 lacks an explicit presentation of the problem formulation, and the input-output structure is not clearly outlined. Additionally, the presence of numerous long sentences makes many sections challenging to follow.

For originality aspect, the idea of recognizing user unique interaction patterns without being influenced by other users with different interaction behaviors has some novelty.

For significance aspect, although HiGSP is proposed for handling a very well-studied problem (top-k recommendation), the performance results of HiGSP are significant compared to existing related methods.

**Questions:**

1. The authors may want to provide a discussion regarding the availability of the six datasets used in experiments. This information is crucial for reproducibility, and to facilitate future research in the field.

**Reviewer Confidence:**

3: The reviewer is confident but not certain that the evaluation is correct

**Scope:**

4: The work is relevant to the Web and to the track, and is of broad interest to the community

---

### Official Review · Reviewer_7wJ1 · 2023-11-24

**Novelty:** 5
**Technical Quality:** 6

**Review:**

The authors propose the Hierarchical Graph Signal Processing (HiGSP) method, a novel approach to collaborative filtering. Inspired by user segmentation theory in marketing, HiGSP strategically segments users into distinct clusters based on shared interaction characteristics. This segmentation should allow for a more nuanced understanding of user preferences, as users within the same cluster are likely to have similar preferences, while those in different clusters exhibit distinct interaction behaviors.

HiGSP employs a two-module approach to filter design. The cluster-wise filter module is designed to recognize user interaction patterns within their respective clusters, ensuring that the recognized patterns are unique and reflective of individual user preferences without external influence. Additionally, the globally-aware filter module complements the cluster-wise module by effectively capturing user general interaction patterns using a high-order low-pass filter.

The authors present extensive experiments conducted on six real-world datasets to evaluate the performance of HiGSP. Comparative analyses against state-of-the-art Graph Convolutional Network (GCN)-based and GSP-based recommendation methods demonstrate the good prediction accuracy of HiGSP.

The paper also provides the source code to replicate the experiments. However, I missed the hyperparameter configurations for the baselines to enable the full reproducibility of the experiments.

Other positive aspects are a time complexity analysis and an extensive set of side studies, including the ablation study.

**Questions:**

I have no questions.

**Ethics Review Description:**

no issues

**Reviewer Confidence:**

3: The reviewer is confident but not certain that the evaluation is correct

**Scope:**

3: The work is somewhat relevant to the Web and to the track, and is of narrow interest to a sub-community

---

### Official Review · Reviewer_BoRw · 2023-11-27

**Novelty:** 4
**Technical Quality:** 5

**Review:**

The research introduces a Hierarchical Graph Signal Processing (HiGSP) method for collaborative filtering, which innovatively employs a dual-module strategy, including a cluster-wise filter module and a globally-aware filter module to enhance recommendation systems' performance by considering both user general behaviors and more personalized behaviors.

Pros
- The paper is well-structured, with a clear introduction to the significance of GSP in recommender systems and a detailed explanation of the proposed HiGSP method.
- The results are promising compared with other GCN-based and GSP-based CF methods.
- The proposed cluster-based filter could benefit the recommendation performance by addressing the individuality of user preferences.

Cons
- The paper does not discuss in depth the method's adaptability to evolving data, which is crucial for maintaining the accuracy of recommendation systems over time.
- The complexity of the HiGSP method might pose challenges in terms of computational resources and execution time, particularly on a larger scale.

**Questions:**

- How does the HiGSP model adapt to changes in user behavior patterns over time?
- Some questions about the clustering approach, what if one user has similar patterns with different users, e.g., the user has both preferences in music and electronic devices, and his/her music preference is similar to user A, and the preference in electronic devices is similar to user B?

**Reviewer Confidence:**

3: The reviewer is confident but not certain that the evaluation is correct

**Scope:**

3: The work is somewhat relevant to the Web and to the track, and is of narrow interest to a sub-community

---

### Decision · Program_Chairs · 2024-01-22

**Decision:**

Accept

**Comment:**

All reviewers agree that this is a well written and motivated paper, proposing an interesting and novel approach to collaborative filtering. The authors have performed an extensive evaluation against SOTA. The main questions have been addressed in the discussion phase and it seems that these are easy to integrate in the final paper. The PC concerns regarding duplicate submissions has also been addressed. We hope that the authors will cite their other work and make the differences clear in the submitted manuscript, as promised.